# Single-Centre Retrospective Evaluation of Intraoperative Hemoadsorption in Left-Sided Acute Infective Endocarditis

**DOI:** 10.3390/jcm11143954

**Published:** 2022-07-07

**Authors:** Jurij Matija Kalisnik, Spela Leiler, Hazem Mamdooh, Janez Zibert, Thomas Bertsch, Ferdinand Aurel Vogt, Erik Bagaev, Matthias Fittkau, Theodor Fischlein

**Affiliations:** 1Department of Cardiac Surgery, Klinikum Nürnberg, Paracelsus Medical University, 90471 Nuremberg, Germany; spela.leiler@klinikum-nuernberg.de (S.L.); hazem.mamdooh@klinikum-nuernberg.de (H.M.); erik.bagaev@klinikum-nuernberg.de (E.B.); matthias.fittkau@klinikum-nuernberg.de (M.F.); theodor.fischlein@klinikum-nuernberg.de (T.F.); 2Medical School, University of Ljubljana, 1000 Ljubljana, Slovenia; 3Department of Biostatistics, Faculty of Health Sciences, University of Ljubljana, 1000 Ljubljana, Slovenia; janez.zibert@zf.uni-lj.si; 4Institute of Clinical Chemistry, Laboratory Medicine and Transfusion Medicine, Paracelsus Medical University, 90471 Nuremberg, Germany; thomas.bertsch@klinikum-nuernberg.de; 5Department of Cardiac Surgery, Artemed Clinic Munich-South, 81379 Munich, Germany; ferdinand.vogt@artemed.de

**Keywords:** infective endocarditis, sepsis, hemoadsorption, CytoSorb, cytokine release syndrome

## Abstract

Background: Cardiac surgery in patients with infective endocarditis (IE) is still associated with high mortality and morbidity; an already present inflammation might further be aggravated due to a cardiopulmonary bypass-induced dysregulated immune response. Intraoperative hemoadsorption therapy may attenuate this septic response. Our objective was therefore to assess the efficacy of intraoperative hemoadsorption in active left-sided native- and prosthetic infective endocarditis. Methods: Consecutive high-risk patients with active left-sided infective endocarditis were enrolled between January 2015 and April 2021. Patients with intraoperative hemoadsorption (Cytosorbents, Princeton, NJ, USA) were compared to patients without hemoadsorption (control). Endpoints were the incidence of postoperative sepsis, sepsis-associated death and in-hospital mortality. Predictors for sepsis-associated mortality and in-hospital mortality were analysed by multivariable logistic regression. Results: A total of 202 patients were included, 135 with active left-sided native and 67 with prosthetic valve infective endocarditis. Ninety-nine patients received intraoperative hemoadsorption and 103 patients did not. Ninety-nine propensity-matched pairs were selected for final analyses. Postoperative sepsis and sepsis-related mortality was reduced in the hemoadsorption group (22.2% vs. 39.4%, *p* = 0.014 and 8.1% vs. 22.2%, *p* = 0.01, respectively). In-hospital mortality tended to be lower in the hemoadsorption group (14.1% vs. 26.3%, *p* = 0.052). Key predictors for sepsis-associated mortality and in-hospital mortality were preoperative inotropic support, lactate-levels 24 h after surgery, C-reactive protein levels on postoperative day 1, chest tube output, cumulative inotropes and white blood cell counts on postoperative day 2, and new onset of dialysis. Multivariate regression analysis revealed intraoperative hemoadsorption to be associated with lower sepsis-associated (OR 0.09, 95% CI 0.013–0.62, *p* = 0.014) as well as in-hospital mortality (OR 0.069, 95% CI 0.006–0.795, *p* = 0.032). Conclusions: Intraoperative hemoadsorption holds promise to reduce sepsis and sepsis-associated mortality after cardiac surgery for active left-sided native and prosthetic valve infective endocarditis.

## 1. Introduction

Infective endocarditis (IE) is associated with significant cardiac and noncardiac morbidity. The growing number of cardiac interventions, valve implantations and staphylococcal infections constantly increases its prevalence [1,2]. Despite significant advancements, IE should not be underestimated, presenting with in-hospital mortality ranging from 20% to sometimes higher than 60% [1,2,3,4,5,6,7].

Surgical complexity, cardiopulmonary bypass (CPB)-induced hyperinflammation and postoperative sepsis have been identified as detrimental factors influencing outcomes [2,3,7]. A deranged immune response may activate a disseminated intravascular coagulation cascade [8,9,10,11]. Furthermore, an interplay of hyperinflammatory mechanisms culminates in end-organ deterioration and disproportionately decreased survival [4,12,13]. Multiorgan failure could be potentially reduced by removing pro-inflammatory circulating cytokines via blood purification, which has been introduced recently into modern cardiac surgery [11,14]. CytoSorb^®^ is a CE-approved cytokine adsorption device with polymer beads that lowers circulating inflammatory mediators and bacterial enterotoxins in the range of up to 60 kDa [11,15]. Intraoperative adsorption demonstrated effective cytokine reduction during IE surgery [16,17], however with no detectable clinical benefit. Other recent studies evaluating intraoperative hemoadsorption during IE surgery reported ambiguous outcomes, from improved hemodynamic stabilization and sepsis attenuation and reduced bleeding with fewer transfusion requirements to increased bleeding diathesis postoperatively [4,5,18,19].

The aim of the present study was (1) to evaluate if intraoperative hemoadsorption could reduce sepsis occurrence or attenuate its severity in patients with active left-sided native valve (NVE) or prosthetic valve endocarditis (PVE) undergoing cardiac surgery, and (2) to evaluate predictors for sepsis, sepsis-associated mortality and in-hospital mortality.

## 2. Materials and Methods

### 2.1. Ethical Statement

The study was conducted in accordance with the Declaration of Helsinki, designed and reported in accordance with STROBE guidelines, [20] registered by the Institutional Study Centre (SZ_W_134.21-I-6) and approved by the Institutional Review Board (IRB-2021–031) on 10 December 2021. Informed consent was waived due to the study’s retrospective design, utilizing routinely obtained de-identified clinical and laboratory data.

### 2.2. Patients

Eligible candidates for this retrospective study were consecutive patients operated for active primarily left-sided NVE and PVE at the Department of Cardiac Surgery, Klinikum Nürnberg, Paracelsus Medical University, Nuremberg, Germany. Between January 2015 and April 2021, 204 patients fulfilling modified Duke criteria were identified [21]. According to guidelines, all patients received antibiotics and anti-infective therapy under supervision of an infectious disease specialist [1,2]. Intraoperative hemoadsorption during CPB was introduced in Nuremberg in 2018 and has been used ever since in most patients with IE. Exclusion criteria were isolated right-sided IE, fungal IE, age < 18 years and video-assisted right lateral approach to the mitral valve. Patients with recurrent IE within one year or identical pathogens before the second surgery were excluded from the analysis [22].

Patients received standardized anaesthesia. After implementing CPB via central cannulation, cardioplegic arrest was induced by cold crystalloid Bretschneider cardioplegia (Custodiol^®^, Dr. F.Koehler Chemie, Bensheim, Germany) or cold blood cardioplegia (Calafiore, custom preparation, Klinikum Nürnberg, Germany). The CytoSorb^®^ (Cytosorbents, Monmouth Junction, NJ, USA) cartridge was installed into the venous system of the CPB between the oxygenator and venous reservoir for the entire duration of CPB. Operative strategies included excessive debridement, valve reconstruction or replacement with potential patch-plasty. Concomitant procedures such as myocardial revascularization, replacement of the ascending aorta, left atrial appendage amputation, or closure of persistent foramen ovale, atrial or ventricular septum defect, were performed whenever indicated.

### 2.3. Data Collection

Patient’s characteristics including demographical and comorbidity-related data, clinical and echocardiographic status including left ventricular ejection fraction, infective agent related informationintraoperative details, and outcomes were retrieved from archived patient files from SAP (Waldorf, Germany) and THG-QIMS (Terraconnect, Nottuln, Germany) quality management software. Laboratory parameters including Hemoglobin, blood lactate, C-reactive protein, white blood cells (WBC) and platelets were determined using the XE-5000 haematology analyser (Sysmex, Norderstedt, Germany), cobas^®^ e602 and c702 module (Roche Diagnostics, Mannheim, Germany) or arterial blood gas analyzer ABL800 (Radiometer, Krefeld, Germany).

Primary outcome was the incidence of sepsis and sepsis-associated mortality. Sepsis was defined by the Third International Consensus Definitions [23]. In-hospital mortality was evaluated as a secondary outcome parameter [24,25]. The rationale of this evaluation was that intraoperative hemoadsorption could reduce sepsis occurrence or attenuate its severity [11,15,26].

### 2.4. Statistical Analysis

The hemoadsorption group was compared with the control group by using unadjusted and propensity scored-matched data. Propensity score matching was performed by first calculating the standardized mean differences (SMD) of the variables, and those variables with SMD > 0.1 were selected for propensity score matching.

Continuous variables were expressed as mean or median with standard deviation (SD) or interquartile range (IQR), respectively, and compared using Student’s t test or the Mann–Whitney test in the case of non-normally distributed data. Categorical data were expressed as the number of patients and frequencies and were compared using the chi-square test. Univariable and multivariable logistic regression analyses were performed to identify independent risk factors for sepsis-associated and in-hospital mortality. Only variables with a *p* value ≤ 0.1 in the univariable analysis were used for the forward stepwise multivariable logistic regression analysis. All statistical analyses were performed by using CRAN R (https://www.R-project.org/, version 3.6, The R Foundation for Statistical Computing, Vienna, Austria, accessed on 10 May 2022).

### 2.5. Definitions

Active IE was defined as an ongoing infection under antibiotic therapy. Coronary heart disease was defined as a history of percutaneous coronary intervention, coronary artery bypass or myocardial infarction. By EuroScore II, the updated system for calculation the risk of death in heart surgery was meant. Liver cirrhosis was defined as cirrhosis of any stage according to Child–Pugh classification. Chronic obstructive pulmonary disease was defined as chronic obstructive pulmonary disease of any stage after GOLD. Re-operation was any surgery due to endocarditis following previous heart surgery. Previous multiple valve surgery was defined as past surgery on two or more valves. Endocarditis of two or more valves was defined when 2 or more valves showed echocardiographic features of endocarditis that were confirmed during surgery. Concomitant right-sided endocarditis was defined as IE of the right heart that occurred secondarily to the endocarditis of the left heart. Concomitant tricuspid valve procedure was every tricuspid valve intervention in addition to the left-sided valve surgery due to endocarditis. Complex surgery was any additional procedure beyond valve repair/replacement, patch reconstruction and/or myocardial revascularization. Major adverse cardiac and cerebrovascular events were defined as per the Society of Thoracic Surgeons for Adult Cardiac Surgery (STS ACS) in-hospital mortality representing the greater of in-hospital, or 30-day mortality, sepsis-associated mortality, myocardial damage, or stroke. Central neurological complications included ischemic events, haemorrhages, cerebral embolisms and abscesses, encephalopathy and meningitis. Sepsis and septic shock were defined by the Third International Consensus Definitions, using quickSOFA criteria [23]. Cardiogenic shock was defined as systolic blood pressure <90 mmHg for >30 min or the need for inotropes to maintain systolic blood pressure >90 mmHg with clinical signs of impaired end-organ perfusion with at least one of the following: cool extremities, decreased urine output, altered mental status, laboratory-confirmed metabolic acidosis, elevated serum lactate and/or creatinine [27]. Postoperative atrial fibrillation (AF) was defined as any new onset of AF that occurred after the surgery during the time of hospitalization. CKD-EPI: the Chronic Kidney Disease Epidemiology Collaboration equation for QFR calculated:141 × min(S_Cr_/κ, 1)^α^ × max(SCr/κ, 1)−1.209 × 0.993 Age in years × 1.018 (if female) × 1.159 (if African American where SCr is standardized serum creatinine)(mg/dL),κ = 0.7 (females) or 0.9 (males),α = −0.329 (females) or −0.411 (males), min = indicates the minimum of SCr/κ or 1, max = indicates the maximum of SCr/κ or 1 [28]. Deep surgical wound infection was defined as sternal osteomyelitis or mediastinitis

## 3. Results

From the initial 204 patients, two were excluded (one patient because IE could not be confirmed and the other patient because he died during surgery). Of the remaining 202, 103 patients were part of the cohort without intraoperative hemoadsorption (operation between January 2015 and December 2017), and the remaining 99 formed the cohort in which intraoperative hemoadsorption was used (operation between January 2018 and April 2021). Fifteen patients from the control and twelve from the hemoadsorption group received an emergent operation. Demographics are summarized in Table 1. Preoperative characteristics were similar in both groups, except for a higher rate of prior cerebrovascular events in the hemoadsorption group (32.3% vs. 18.4%, *p* = 0.035, Table 1). Thus, propensity score matching was performed to mitigate the potential confounding effects yielding 99 pairs with similar baseline preoperative and operative characteristics as presented in the right columns of Table 1.

### 3.1. Operative Characteristics

As depicted in Table 2, the median interval between definitive diagnosis for surgical indication and surgery was 4 days (range 2–9 days) with no difference between both groups. In addition, both groups did not show any difference in the overall time of preoperative anti-infective treatment and were comparable with regard to operative characteristics both in non-adjusted and matched cohort. No device-related adverse events in the hemoadsorption group occurred, and none of the patients received postoperative hemoadsorption therapy.

### 3.2. Outcomes

All outcome parameters are summarized in Table 3, demonstrating consistent results for unadjusted and propensity-matched cohorts. Therefore, only propensity-matched data are referred to further in the text. The incidence of postoperative sepsis was 22.2% in the hemoadsorption compared to 39.4% (*p* = 0.014) in the control group. Sepsis-associated mortality also differed between both groups (8.1% vs. 22.2%, *p* = 0.010) (Figure 1, Table 3). Overall, in-hospital mortality was notably but insignificantly lower in the hemoadsorption group (14.1% vs. 26.3, *p* = 0.052). Lower C-reactive protein (CRP) levels on the first postoperative day (8.8 vs. 11.1 mg/dL; *p* = 0.024) and lower leukocyte counts on the second postoperative day (9.9 vs. 12.1 × 10^3^/µL; *p* = 0.021) were found in the hemoadsorption group (Figure 2).

### 3.3. Regression Analyses for Sepsis, Sepsis-Associated Mortality and In-Hospital Mortality

Several parameters predicted sepsis, sepsis-associated and in-hospital mortality, as summarized in Table 4, Table 5 and Table 6. Intraoperative hemoadsorption was an independent preventive factor both for sepsis-associated and in-hospital mortality (*p* = 0.014 and *p* = 0.032, respectively). Postoperative renal replacement therapy was independently associated with sepsis (*p* = 0.003), sepsis-associated (*p* = 0.015) and in-hospital mortality (*p* = 0.018). C-reactive protein after 24 h and white blood cell counts after 48 h were independent predictors of both sepsis (*p* = 0.021 and *p* < 0.001) and sepsis-associated mortality (*p* = 0.024 and *p* = 0.006, respectively). An additional independent predictor of sepsis was Staphylococcus species infection (*p* = 0.041).

Cumulative inotropes after 48 h were independent risk factors both for sepsis-associated and in-hospital mortality (*p* = 0.015 and *p* = 0.024, respectively). Preoperative levels of inotropes, lactate levels and chest tube output 24 h after surgery were identified as further independent risk factors for in-hospital mortality (*p* = 0.014, *p* = 0.049, and *p* = 0.002, respectively).

## 4. Discussion

The current study compared adjunctive intraoperative hemoadsorption therapy versus cardiac surgery without hemoadsorption in a cohort of consecutive high-risk patients presenting with active prosthetic or native valve left-sided IE. The following main observations can be inferred from the current study. First, the incidence of sepsis and sepsis-associated mortality was lower in the hemoadsorption group. The observed reduction corresponded to lower CRP levels 24 h after surgery and lower WBC counts on the second postoperative day. Second, a relevant however insignificant difference in overall in-hospital mortality was observed, with paralleled higher haemoglobin levels on the first postoperative day and lower RBC transfusion requirements in the hemoadsorption group. Third, multifactorial analyses revealed the association of intraoperative hemoadsorption with reduced sepsis-associated and in-hospital mortality. Finally, intraoperative hemoadsorption was safe and did not show any device-related events.

According to the European Infective Endocarditis Registry data, two thirds of in-hospital mortality could be attributed to non-cardiovascular or a combination of cardiovascular and non-cardiovascular causes, whereby sepsis accounted for 75% of non-cardiovascular causes (18). Therefore, a minority of in-hospital mortality is regarded as sepsis-independent. Especially in the surgical treatment of IE, outcomes were influenced mainly by surgical complexity, preoperative patient conditions and comorbidities, as well as the deleterious effects of prolonged CPB and aortic cross clamp (ACC) times [12,13,24]

Despite many advancements in modern anti-infective treatment strategies, IE remains a potentially fatal disorder, and currently, controversies about therapy, including medical and surgical approaches, persist. Anticipating an association between the infection-induced inflammatory response and IE outcomes, the profound understanding of underlying mechanisms including dysregulated immune response to IE-related triggers has become an important target of recent investigations [11,14,18,29,30,31].

Specifically, the recognition of biofilm formation coming with increased antibiotic resistance and hyperinflammatory host response amenable to blood purification therapies are subjects of great interest [11,15,16,30,31]. In the treatment armamentarium of patients presenting with sepsis, various blood purification techniques have been investigated thus far, and most recently, these new adjunctive therapies are also being investigated in infective endocarditis patients.

A first study reporting reduced inotropic support, sepsis and sepsis-associated mortality in patients with native mitral valve IE coupled with hemoadsorption therapy was published by Haidari et al. [18]

Holmén confirmed these results in a small-randomized control trial in patients requiring urgent surgery for IE where hemoadsorption therapy was used intraoperatively. Not only was the accumulated dose of inotropes double in the control group, there was a significantly lower need for blood products in the treatment group [19]. However, another study published by Santer et al. showed different results without beneficial effects of hemoadsorption therapy in IE patients compared to Haidari et al. [5]. More recently, the REMOVE (Revealing Mechanisms and Investigating Efficacy of Hemoadsorption for Prevention of Vasodilatory Shock in Cardiac Surgery Patients with Infective Endocarditis) randomized controlled trial was published, showing no difference in the primary endpoint of the delta (Δ) Sequential Organ Failure Assessment (SOFA) score [17]. In contrast to REMOVE, in the present analysis, we focused on sepsis and sepsis-related mortality, rather than on the ΔSOFA score, as the SOFA score has not been validated for cardio-surgical patients. Moreover, in the present study, over one third of patients from the overall cohort were on preoperative inotropic support; 16.2% in the control and 19.2% in the hemoadsorption group (*p* = 0.709) presented with septic shock (≈80% in NYHA class III/IV). Specifically, this means that in the present analysis, all-comer and consecutive patients were enrolled to reflect “real-world” data, compared to REMOVE, where only 288 out of 740 screened patients were recruited [17].

As the major finding, we observed significantly reduced sepsis-associated mortality and milder postoperative hyperinflammation. Whether this finding could translate into better overall survival is, however, debatable, but the present analysis demonstrated a relevant difference in overall in-hospital mortality without reaching statistical significance (26.3% vs. 14.1%, *p* = 0.052). Among other factors, we could show for the first time that intraoperative hemoadsorption is associated with lower sepsis-associated (OR 0.091, 95% CI 0.013–0.620, *p* = 0.014) and in-hospital mortality (OR 0.069, 95% CI 0.0006–0.795, *p* = 0.032) in IE patients. If it is postulated that intraoperative hemoadsorption could result in less sepsis and sepsis-associated mortality, the treatment time may be considered a limitation, and a longer duration of hemoadsorption may exert additional effects as also mentioned by the REMOVE investigators. In all published trials conducted thus far regarding evaluating the effect of hemoadsorption in IE, the device was only used during the index procedure. We are only at the beginning of a better understanding of the adjunctive potential of hemoadsorption therapy in IE patients. The REMOVE trial showed a significant reduction in all plasma-circulating harmful cytokines including cell-free DNA in the hemoadsorption group. Moreover, within the REMOVE trial, the direct correlation between CPB-time and elevation of interleukin (IL)-6 was demonstrated. Innovative markers such as pro-adrenomedullin, as mentioned in REMOVE, might serve for better patient selection in the future [17].

### Limitations

The present study comes with the following limitations: The study was not a randomized controlled trial, but both groups were comparable. Nonetheless, bias cannot be completely excluded. Moreover, although the current trial represents the largest single-centre cohort of IE patients treated by hemoadsorption thus far, the sample size is still too small to draw definitive conclusions. Due to the overall recruitment period of 6 years, surgical techniques and antimicrobial therapy have changed slightly, opening the study to treatment biases. Second, this was a single-centre, retrospective database analysis, whereby special advanced clinical and inflammatory parameters (e.g., systemic vascular resistance, IL-6, Procalcitonin, N-terminal prohormone of brain natriuretic peptide) were not available for meaningful analyses due to missing values. Further, the retrospective design precluded routine determination of coagulation and endogenous vasoconstricting factors, such as protein C, antithrombin III, coagulation factors VII and X, cortisol, thromboxane, endothelin-1 and albumin.

## 5. Conclusions

As the main finding, the present analysis demonstrates a reduction in sepsis and sepsis-associated mortality by the intraoperative use of hemoadsorption with CytoSorb^®^ in high-risk IE patients with affected left-sided native or prosthetic valves. Intraoperative hemoadsorption was safe and easy to use without any adjustment to the intraoperative heparin regime. Excessive postoperative inflammatory response with the increased need of postoperative inotropic support and kidney failure requiring dialysis were independent risk factors for sepsis-associated mortality. The intraoperative use of CytoSorb^®^ was the only preventive factor in this regard. Preoperative inotropes, elevated lactate levels at 24 h after the surgery, kidney failure requiring dialysis and excessive postoperative bleeding were independently associated with in-hospital mortality. Again, the intraoperative use of CytoSorb^®^ was a preventive factor. More evidence is needed to better define the value of hemoadsorption in cardiac surgery, especially in the setting of IE relating to appropriate patient selection, timing and dosing.

## Figures and Tables

**Figure 1 jcm-11-03954-f001:**
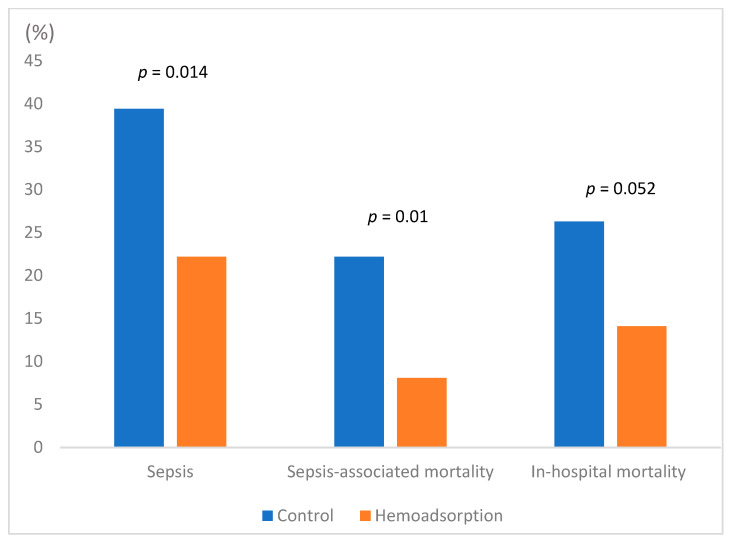
Sepsis, sepsis-associated mortality, and in-hospital mortality (propensity-matched cohort).

**Figure 2 jcm-11-03954-f002:**
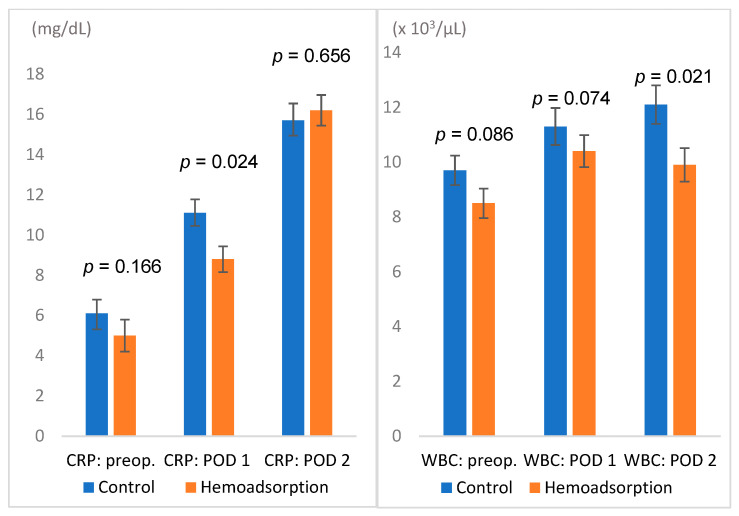
Basic laboratory values: C-reactive protein (mg/dL) and white blood cell counts (×10^3^/µL); POD: postoperative day; WBC: white blood cell count.

**Table 1 jcm-11-03954-t001:** Preoperative patient characteristics.

Preoperative Characteristics						
	Unadjusted			Propensity Score Match		
	Control (*n* = 103)	Hemoadsorption (*n* = 99)	*p* Value	Control (*n* = 99)	Hemoadsorption (*n* = 99)	*p* Value
Demographics						
Age (years)	69 [58;77]	67 [58;75]	0.612	68.0 [56.5;76.0]	68.0 [56.5;76.0]	0.745
BMI (kg/m^2^)	26.4 [23.8;30.8]	26.8 [24.0;30.6]	0.891	26.4 [23.9;31.0]	26.8 [24.0;30.6]	0.951
Gender (% male)	83 (80.6%)	81 (81.8%)	0.964	79 (79.8%)	81 (81.8%)	0.857
EuroScore II (%)	9 [3.6;22.2]	9.9 [5.5;21.8]	0.805	8.95 [3.58;21.1]	9.89 [5.5;21.8]	0.705
Native valve IE	76 (37.6%)	49 (24.1%)	0.106	73 (73.7%)	49 (24.1%)	0.110
Prosthetic valve IE	27 (13.3%)	40 (19.9%)	0.169	27 (27.3%)	40 (19.9%)	0.218
CAD	28 (27.5%)	28 (28.3%)	1.000	25 (25.5%)	28 (28.3%)	0.781
Diabetes Mellitus (II)	25 (24.3%)	23 (23.2%)	0.993	25 (25.3%)	23 (23.2%)	0.868
pAOD	6 (5.8%)	3 (3%)	0.499	4 (4.04%)	3 (3.03%)	1.000
History of CVI	19 (18.4%)	32 (32.3%)	0.035	19 (19.2%)	32 (32.3%)	0.051
Septic embolisms (last 3 weeks)	19 (18.4%)	21 (21.2%)	0.752	19 (19.2%)	21 (21.2%)	0.860
COPD	20 (19.4%)	14 (14.1%)	0.416	18 (18.2%)	14 (14.1%)	0.562
Liver cirrhosis	4 (3.9%)	6 (6.1%)	0.533	4 (4.08%)	6 (6.1%)	0.747
AF preop.	20 (19.4%)	27 (27.3%)	0.248	20 (20.2%)	27 (27.3%)	0.316
eGFR_CKD-EPI_ (mL/min)	62 [38;79]	52 [40;74]	0.380	62.0 [39.5;78.5]	52.0 [40.0;74.0]	0.340
RRT preop.	10 (9.7%)	8 (8.1%)	0.874	9 (9.09%)	8 (8.1%)	1.000
Antiplatelet therapy	32 (31.1%)	22 (23.2%)	0.276	30 (30.3%)	22 (23.2%)	0.337
Oral anticoagulant therapy	13 (12.7%)	10 (10.4%)	0.772	12 (12.2%)	10 (10.4%)	0.861
Clinical status						
NYHA (III-IV)	73 (70.9%)	78 (78.8%)	0.257	72 (72.7%)	78 (78.8%)	0.407
Inotropes preop.	36 (35.0%)	40 (40.8%)	0.477	35 (35.4%)	40 (40.8%)	0.520
Septic shock (within 48 h)	17 (16.5%)	19 (19.2%)	0.753	16 (16.2%)	19 (19.2%)	0.709
Ventilated preop.	12 (11.7%)	9 (9.2%)	0.733	12 (12.1%)	9 (9.2%)	0.662
Laboratory Parameters						
CRP (mg/dL) preop.	6 [3;10.9]	5 [2;10.3]	0.166	6.1 [3.00;11.0]	5.0 [2.0;10.3]	0.146
Platelets (×10^3^/µL) preop.	236 [160;307]	238 [171;300]	0.920	237 [157;314]	238 [171;300]	0.983
WBC (×10^3^/µL) preop.	9.7 [7.3;14.1]	8.5 [6.2;12.5]	0.077	9.30 [7.25;14.4]	8.50 [6.2;12.5]	0.086
Hb (g/dL) preop.	10.2 [9.1;11.4]	10.1 [9.1;11.5]	0.836	10.3 [9.15;11.5]	10.1 [9.1;11.5]	0.999
Lactate (mmol/l) at start of surgery	0.8 [0.6;1.1]	0.7 [0.5;1]	0.183	0.80 [0.60;1.10]	0.70 [0.5;1.0]	0.199
Echocardiographic/Radiologic Characteristics						
LV-EF lower than 50%	29 (28.2%)	23 (23.2%)	0.523	25 (25.3%)	23 (23.2%)	0.868
Vegetations	98 (95.1%)	94 (94.9%)	1.000	94 (94.9%)	94 (94.9%)	1.000
Paravalvular extension or abscess	37 (35.9%)	44 (44.4%)	0.275	36 (36.4%)	44 (44.4%)	0.311
Concomitant right-sided endocarditis	5 (4.9%)	2 (2.1%)	0.446	5 (5.05%)	2 (2.1%)	0.445
Causative infective agent						
Staphylococcus spec.	40 (38.8%)	31 (32.0%)	0.385	40 (40.4%)	31 (32.0%)	0.280
Staphylococcus aureus	26 (26%)	19 (21.3%)	0.563	26 (27.1%)	19 (21.3%)	0.461
Streptococcus species	29 (29%)	18 (20.2%)	0.221	26 (27.1%)	18 (20.2%)	0.357
Enterococcus faecalis	13 (13%)	15 (16.9%)	0.590	13 (13.5%)	15 (16.9%)	0.672
Gram-bacteria	5 (4.9%)	5 (5.1%)	1.000	5 (5.10%)	5 (5.1%)	1.000
Antibiotic therapy (d) preop.	6 [2;11]	5 [3;10.8]	0.702	6 [2;11]	5 [3;11]	0.850

Data are presented as median [interquartile range], mean (standard deviation), or *n* (%). AF: atrial fibrillation; BMI: body mass index; CAD: coronary artery disease; CKD-EPI: Chronic Kidney Disease Epidemiology Collaboration—for the equation see Section 2; CK-MB: Creatine kinase-MB; c-TnT: cardiac troponin; CRP: C-reactive protein; CVI: cerebrovascular insult; COPD: chronic obstructive pulmonary disease; eGFR: estimated glomerular filtration rate; EuroScore II: updated system for calculation the risk of death in heart surgery; h: hour; Hb: haemoglobin; IE: infective endocarditis; Liver cirrhosis: cirrhosis of any stage according to Child-Pugh classification; LV-EF: left ventricular ejection fraction determined by echocardiography; NYHA: New York Heart Association Classification; oral anticoagulant therapy: Vitamin K antagonists or new oral anticoagulants; pAOD: peripheral arterial obstructive disease; preop.: preoperatively; RRT: renal replacement therapy; inotropes: (nor-) +epinephrine + dobutamine; WBC: white blood cells.

**Table 2 jcm-11-03954-t002:** Intraoperative data.

Intraoperative Data						
	Unadjusted			Propensity Score Match		
	Control (*n* = 103)	Hemoadsorption (*n* = 99)	*p* Value	Control (*n* = 99)	Hemoadsorption (*n* = 99)	*p* Value
Time: diagnosis and surgery (d)	4 [2;8.3]	4 [2;9.8]	0.697	4.00 [2.0;7.3]	4.00 [2.0;9.8]	0.546
Re-operation	43 (41.8%)	46 (46.5%)	0.769	43 (43.4%)	46 (46.5%)	0.891
Cardiopulmonary Bypass (min)	126 [93.0;168]	134 [108;176]	0.212	126 [93.0;168]	134 [108;176]	0.204
Aortic cross-clamp (min)	82 [59;116]	91 [72.5;131]	0.063	82.0 [58.0;114]	91.0 [72.5;131]	0.058
Selective cerebral perfusion	2 (1.9%)	4 (4%)	0.661	2 (2%)	4 (4%)	0.697
Isolated aortic valve surgery	56 (54.4%)	46 (46.5%)	0.601	56 (56.6%)	46 (46.5%)	0.833
Aortic bioprosthesis	56 (54.4%)	50 (50.5%)	0.852	50 (50.5%)	46 (46.5%)	0.833
Aortic mechanical prosthesis	8 (7.8%)	9 (9.1%)	0.953	8 (8.1%)	9 (9.1%)	1.000
Complex aortic surgery	14 (13.6%)	23 (23.2%)	0.196	10 (10.1%)	23 (23.2%)	0.056
Aortic and mitral valve surgery	9 (8.7%)	12 (12.1%)	0.631	14 (14.3%)	8 (8.1%)	0.248
Isolated mitral valve surgery	29 (28.2%)	27 (27.3%)	1.000	25 (25.3%)	28 (28.3%)	0.748
Mitral bioprosthesis	25 (24.3%)	26 (26.3%)	0.925	21 (21.2%)	22 (22.2%)	1.000
Mitral mechanical prosthesis	8 (7.8%)	7 (7.1%)	1.000	8 (8.1%)	7 (7.1%)	1.000
Mitral valve reconstruction	5 (4.9%)	6 (6.1%)	0.962	8 (8.1%)	5 (5.1%)	1.000
Concomitant tricuspid valve surgery	4 (3.9%)	6 (6.1%)	0.533	4 (4.08%)	6 (6.1%)	0.747
Concomitant revascularization	14 (14.0%)	14 (14.6%)	1.000	12 (12.1%)	14 (14.1%)	0.833
Pericardial patch reconstruction	36 (35.6%)	36 (36.4%)	1.000	36 (37.1%)	36 (36.4%)	1.000

Data are presented as median [interquartile range], mean (standard deviation) or *n* (%), d: day; min.: minute.

**Table 3 jcm-11-03954-t003:** Postoperative outcomes.

Postoperative Outcomes						
	Unadjusted Data			Propensity Score Matched Data		
	Control (*n* = 103)	Hemoadsorption (*n* = 99)	*p* Value	Control (*n* = 99)	Hemoadsorption (*n* = 99)	*p* Value
Sepsis postop.	39 (37.9%)	22 (22.2%)	0.023	39 (39.4%)	22 (22.2%)	0.014
Sepsis-associated mortality	22 (21.4%)	8 (8.1%)	0.014	22 (22.2%)	8 (8.1%)	0.010
In-hospital mortality	26 (25.2%)	14 (14.1%)	0.071	26 (26.3%)	14 (14.1%)	0.052
Re-thoracotomy for bleeding	17 (16.7%)	10 (10.2%)	0.258	17 (17.3%)	10 (10.2%)	0.214
Mechanical ventilation (h)	22 [8, 68.2]	19.0 [10, 65.5]	0.856	22.0 [7.25, 72.0]	19.0 [10.0, 65.5]	0.813
Chest tube output (ml) 24 h postop.	500 [300, 700]	500 [250, 950]	0.761	500 [300, 700]	500 [250, 950]	0.794
New RRT postop.	25 (24.3%)	19 (19.2%)	0.481	24 (24.2%)	19 (19.2%)	0.491
Central neurological complications	2 (1.9%)	8 (8.1%)	0.055	2 (2.02%)	8 (8.1%)	0.105
Pneumonia	10 (9.7%)	6 (6.1%)	0.484	10 (10.1%)	6 (6.1%)	0.434
Deep Sternal Wound Infection	1 (1.0%)	0	1.000	1 (1.01%)	0 (0.0%)	1.000
UTI	2 (1.9%)	1 (1.0%)	1.000	2 (2.02%)	1 (1.0%)	1.000
Cumulative inotropes (mg) POD1	12.6 [4.9, 40.1]	17.8 [7.8, 37.2]	0.204	12.6 [4.85, 40.1]	17.8 [7.8, 37.2]	0.227
Cumulative inotropes (mg) POD2	5.8 [0.9, 15.2]	4.9 [1.1, 16.9]	0.927	5.82 [0.86, 16.2]	4.93 [1.1, 16.9]	0.877
Lactate (mmol/L) at the end of Surgery	1.3 [1, 1.7]	1.2 [0.9, 1.8]	0.656	1.30 [1.00, 1.70]	1.20 [0., 1.80]	0.652
Lactate (mmol/L) POD1	1.20 [0.90, 1.60]	1.30 [0.92, 1.75]	0.191	1.20 [0.90, 1.60]	1.30 [0.9, 1.8]	0.148
Lactate (mmol/L) POD2	1 [0.7, 1.2]	1 [0.8, 1.3]	0.142	1 [0.7, 1.2]	1 [0.8, 1.3]	0.191
WBC (×10^3^/µL) POD1	11.4 [9.10, 17.1]	10.4 [8.10, 15.6]	0.065	11.3 [9.05, 17.1]	10.4 [8.1, 15.6]	0.074
WBC (×10^3^/µL) POD2	12.1 [8.75, 15.1]	9.90 [7.40, 14.2]	0.025	12.1 [8.90, 15.3]	9.90 [7.4, 14.2]	0.021
CRP (mg/dL) POD1	10.0 [7.2, 15.4]	8.80 [5.0, 12.9]	0.026	11.1 [7.0, 15.8]	8.8 [5.0, 12.9]	0.024
CRP (mg/dL) POD2	15.4 [11.5, 21.8]	16.2 [11, 20.6]	0.824	15.7 [11.5, 22.3]	16.2 [11, 20.6]	0.656
Hb (g/dL) POD1	9.20 [8.4, 9.9]	9.70 [8.7, 10.3]	0.011	9.2 [8.4, 9.9	9.70 [8.7, 10.3]	0.012
Hb (g/dL) POD2	8.75 [8.1, 9.3]	8.90 [8.4, 9.6]	0.166	8.8 [8.1, 9.3]	8.90 [8.4, 9.6]	0.200
Platelets transfused (Units)	0 [0, 2]	0 [0, 1]	0.719	0 [0, 1.5]	0 [0, 1]	0.801
RBCs transfused	3.0 [1.0, 6.0]	1.0 [0.0, 4.0]	0.016	3.0 [1.0, 6.0]	1.0 [0.0, 4.0]	0.016
FFPs transfused	0.0 [0.0, 4.00]	0.0 [00, 6.00]	0.014	0.0 [0.0, 4.0]	0.0 [0.0, 6.0]	0.023
Intensive Care Unit stay (d)	3 [2, 6]	4 [2, 8.5]	0.189	3 [2, 6]	4 [2, 8.5]	0.238
Hospital stay (d)	34.5 [24, 46]	39 [22, 49.2]	0.312	33.5 [24, 46]	39 [22, 49]	0.282

Data are presented as median [interquartile range], mean (standard deviation) or *n* (%), Central neurological complications: ischemic and haemorrhagic events, encephalopathy, meningitis and brain abscess-transient or permanent; cumulative inotropes: (nor--/ + epinephrine-/ + dobutamine; d: day; deep surgical wound infection: including osteomyelitis or mediastinitis; FFP: fresh frozen plasma; Hb: haemoglobin; h: hour; RBC: red blood cell; RRT: renal replacement therapy; POD: postoperative day; postop: postoperatively; inotropes: (nor-) + epinephrine + dobutamine; UTI: urinary tract infection: including urethritis, cystitis and pyelonephritis WBC: white blood cells.

**Table 4 jcm-11-03954-t004:** Variables related to sepsis.

Variables Related to Sepsis (Matched Cohort)				
	Univariate Analysis		Multivariate Analysis	
Variable	OR (Cl_L_, CL_U_)	*p* Value	OR (Cl_L_, CL_U_)	*p* Value
BMI (m^2^/kg)	1.098 (1.026, 1.174)	0.007		
Hemoadsorption	0.308 (0.130, 0.730)	0.008	0.42 (0.16, 1.07)	0.07
Euroscore II	1.022 (1.003, 1.041)	0.024		
Cardiogenic shock in last 48 h	6.125 (2.616, 14.339)	<0.001		
Ventilated preop.	3.326 (1.214, 9.110)	0.019		
Staphylococcus spec.	2.506 (1.126, 5.577)	0.024	2.65 (1.04, 6.75)	0.041
CRP preop.	1.104 (1.054, 1.158)	<0.001		
WBC preop.	1.18 (1.046, 1.196)	0.001		
Lactate preop.	1.878 (1.114, 3.165)	0.018		
Inotropes preop.	7.266 (2.933, 18.002)	<0.001		
CPB duration	1.006 (1.001, 1.011)	0.011		
Aortic cros-clamp	1.007 (1.000, 1.014)	0.046		
Lactate end of surgery	2.207 (1.519, 3.209)	<0.001		
Inotropes cumulative POD 1	1.047 (1.031, 1.065)	<0.001		
Inotropes cumulative POD2	1.05 (1.030, 1.071)	<0.001		
WBC POD1	1.222 (1.131, 1.320)	<0.001		
WBC POD2	1.258 (1.155, 1.370)	<0.001	1.28 (1.15, 1.43)	<0.001
Lactate 24 h postop.	2.644 (1.650, 4.236)	<0.001		
c-TnT POD1	1.351 (1.162, 1.572)	<0.001		
CK-MB POD 1	1.01 (1.004, 1.016)	<0.001		
CRP POD 1	1.088 (1.025, 1.156)	0.006		
CRP POD2	1.056 (0.998, 1.116)	0.059	1.07 (1.01, 1.14)	0.021
New RRT postop.	7.142 (3.101, 16.452)	<0.001	4.93 (1.70, 14.26)	0.003
IABP postop.	6.308 (1.485, 26.792)	0.013		
FFPs transfused	1.088 (1.020, 1.161)	0.011		

Multivariate analysis was performed for *p* < 0.05; BMI: body mass index; CK-MB: creatin kinase MB; CPB: cardiopulmonary bypass; CRP: C-reactive protein; c-TnT: c-troponin; Inotropes: (nor)-/ + epinephrine-/ + dobutamine; FFP: fresh frozen plasma; IABP: intra-aortic balloon pump; min: minute; preop: preoperative; postop: postoperative; POD: postoperative day; RRT: renal replacement therapy, WBC: white blood cells.

**Table 5 jcm-11-03954-t005:** Variables related to sepsis-associated mortality.

Variables Related to Sepsis-Associated Mortality (Matched Cohort)				
	Univariate Analysis		Multivariate Analysis	
Variable	OR (Cl_L_, CL_U_)	*p* Value	OR (Cl_L_, CL_U_)	*p* Value
BMI	1.098 (1.026, 1.174)	0.007		
Hemoadsorption	0.308 (0.130, 0.730)	0.008	0.09 (0.01, 0.62)	0.014
EuroScore II	1.022 (1.003, 1.041)	0.024		
*Staphylococcus spec.*	2.506 (1.126, 5.577)	0.024		
Inotropes preop.	7.266 (2.933, 18.002)	<0.001		
CRP preop.	1.104 (1.054, 1.158)	<0.001		
Cardiogenic shock in the last 48 h	6.125 (2.616, 14.339)	<0.001		
Ventilated preop.	3.326 (1.214, 9.110)	0.019		
WBC preop.	1.118 (1.046, 1.196)	0.001		
Lactate: start of surgery	1.878 (1.114, 3.165)	0.018		
Concomitant tricuspid valve surgery	4.32 (1.139, 16.390)	0.031		
CPB duration (min.)	1.006 (1.001, 1.011)	0.011		
Aortic cross-clamp (min.)	1.007 (1.000, 1.014)	0.046		
Paravalvular extension or Abscess	2.179 (0.992, 4.787)	0.052	4.18 (0.73,23.99)	0.109
Lactate (mmol/L) end of surgery	2.207 (1.519, 3.209)	<0.001		
WBC POD1	1.222 (1.131, 1.320)	<0.001		
WBC POD2	1.258 (1.155, 1.370)	<0.001	1.21 (1.06, 1.38)	0.006
Lactate at 24 h postop.	2.644 (1.650, 4.236)	<0.001		
Inotropes cumulative POD1	1.047 (1.030, 1.064)	<0.001	1.03 (0.99, 1.06)	0.068
Inotropes cumulative POD2	1.005 (1.002, 1.007)	<0.001	1.01 (1.00, 1.01)	0.015
CRP POD 1	1.088 (1.025, 1.156)	0.006	1.15 (1.02, 1.31)	0.024
CRP POD 2	1.056 (0.998, 1.116)	0.059		
c-TnT POD 1	1.351 (1.162, 1.572)	<0.001		
CK-MB POD 1	1.01 (1.004, 1.016)	0.001		
New RRT postop.	7.142 (3.101, 16.452)	<0.001	8.16 (1.49, 44.37)	0.015
FFPs transfused	1.088 (1.020, 1.161)	0.011		
IABP postop.	6.308 (1.485, 26.792)	0.013		

Multivariate analysis was performed for *p* < 0.05; BMI: body mass index; CK-MB: creatin kinase MB; CPB: Cardiopulmonary Bypass; CRP: C-reactive protein; c-TnT: c-troponin; Inotropes: (nor)-/ + epinephrine-/ + dobutamine; FFP: fresh frozen plasma; IABP: intra-aortic balloon pump; min: minute; preop: preoperative; postop: postoperative; POD: postoperative day; RRT: renal replacement therapy, WBC: White blood cells.

**Table 6 jcm-11-03954-t006:** Variables related to in-hospital mortality.

	Univariate Analysis		Multivariate Analysis	
Variable	OR (Cl_L_, CL_U_)	*p* Value	OR (Cl_L_, CL_U_)	*p* Value
BMI	1.068 (1.005, 1.135)	0.034		
Hemoadsorption therapy	0.462 (0.225, 0.951)	0.036	0.07 (0.01, 0.79)	0.032
Staphylococcus spec.	2.497 (1.223, 5.096)	0.012		
Endocarditis of 2 or more valves	2.654 (1.025, 6.873)	0.044		
Cardiogenic chock in the last 48 h	6.786 (3.047, 15.114)	<0.001		
Inotropes preop.	6.362 (2.932, 13.801)	<0.001	17.36 (1.77, 170.13)	0.014
Preop Lactate	2.147 (1.252, 3.679)	0.005		
Ventilated preop.	5.576 (2.168, 14.338)	<0.001		
WBC preop.	1.129 (1.058, 1.204)	<0.001		
CRP preop.	1.082 (1.036, 1.130)	<0.001		
CPB duration	1.006 (1.002, 1.011)	0.008		
Aortic cross-clamp	1.007 (1.001, 1.014)	0.026		
Lactate end of surgery	2.637 (1.761, 3.948)	<0.001		
Lactate 24 h postop.	2.754 (1.712, 4.432)	<0.001	4.14 (1.01, 17.01)	0.049
WBC POD1	1.173 (1.100,1.252)	<0.001		
Inotropes cumulative POD1	1.053 (1.035, 1.071)	<0.001		
Inotropes cumulative POD2	1.048 (1.028, 1.069)	<0.001	1.02 (1.00, 1.04)	0.024
WBC POD2	1.202 (1.120, 1.289)	<0.001		
c-TnT POD 1	1.365 (1.168, 1.595)	<0.001		
CK-MB POD 1	1.009 (1.003, 1.015)	0.003		
eGFR_CKD-EPI_ preop.	0.984 (0.971, 0.998)	0.022		
New RRT postop.	6.833 (3.174, 14.709)	<0.001	14.10 (1.57, 126.67)	0.018
FFPs transfused	1.151 (1.076, 1.230)	<0.001		
RBCs transfused	1.085 (1.025, 1.149)	0.005	0.9 (0.78, 1.03)	0.123
Drainage Output (ml) in 24 h	1.001 (1.000, 1.001)	0.004	1.004 (1.001, 1.006)	0.002
IABP postop.	4.278 (1.021, 17.922)	0.047		

Multivariate analysis was performed for *p* < 0.05; BMI: body mass index; CKD-EPI: Chronic Kidney Disease Epidemiology Collaboration—for the equation see Definitions; CK-MB: creatin kinase MB; CRP: C-reactive protein; c-TnT: c-troponin; Inotropes: (nor)-/ + epinephrine-/ + dobutamine; eGFR: estimated glomerular filtration rate; FFP: fresh frozen plasma; IABP: intra-aortic balloon pump; min: minute; preop: preoperative; postop: postoperative; POD: postoperative day; RBC: red blood cell concentrate; RRT: renal replacement therapy; WBC: White blood cells.

## Data Availability

The data presented in this study are available on request from the corresponding author. The data are primarily not publicly available due to the data protection policy of the institution.

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
