# Peer review of "Single-Centre Retrospective Evaluation of Intraoperative Hemoadsorption in Left-Sided Acute Infective Endocarditis"

_jcm, 2022, doi:10.3390/jcm11143954_

Round 1

Reviewer 1 Report

Regarding the manuscript entitled: "Intraoperative hemoadsorption in left-sided infective endocarditis", I would like to congratulate the authors for the manuscript and the interesting research.

Introduction:

-        Authors should focus the introduction on the previous literature regarding the use of hemoadsorption the intraoperative period. In my opinion, potential readers of the manuscript are probably already aware of the seriousness of infective endocarditis, but may not be aware of the usefulness of hemoadsorption devices demonstrated to date. Therefore, I recommend extending the information provided about these devices in the second paragraph of the introduction. The authors should point out if there is already literature on the intraoperative use of these devices. If this were not the case, I think it is also appropriate to point it out, since it would increase the novelty of the manuscript. Ç

-        In my opinion, the objectives are not well set. After reading the manuscript, I understand that the main objective is to demonstrate the usefulness of "intraoperative hemoadsorption in reducing complications in patients with IE".

-        At the end of the introduction, it should be included the sentence located on page 3 lines 96 - 98: "The rationale of this evaluation was that intraoperative haemadsorption could reduce sepsis occurrence or attenuate its severity". I think it is a good summary of the final objective of the presented study.

Materials and Methods:

-        In my opinion, the information regarding the “Ethical statement” should be placed in the first paragraph of this section. It should be specified the trial registration (if any) and the ethics committee acceptance code should be specified with the approval date.

-        Authors should also include the STROBE checklist to report observational studies. In case they had not followed this checklist, I recommend modifying the manuscript to fit the STROBE checklist.

-        The authors note that "patient's characteristics and risk factors were retrieved." I think they should specify (at least briefly) what characteristics and risk factors were collected to carry out the analysis. They should also specify in this section which "laboratory measurements" were included.

Results:

-        In my opinion, the inclusion of patients could be expressed more clearly. It is not necessary to point out that they were patients submitted for IE (since it was the inclusion criterion).

-        It could be pointed out more clearly that of the 204 patients initially included, 2 were excluded (one patient because IE could not be confirmed and the other patient because he died during surgery). Of the remaining 202, 103 patients were part of the cohort without intraoperative hemoadsorption (operated between January 2015 and December 2017) and the remaining 99 formed the cohort in which intraoperative hemoadsorption was used (operated between January 2018 and April 2021).

-        I do not see it necessary to make redundant in the text the information provided by the tables "(32.3% vs, 18.4%, P=0.035)" (page 3, lines 117 – 118).

-        Table 1: The title should be more concise: "Preoperative patient characteristics". I think also that it is enough to provide comparative information between both cohorts. It is not necessary to give the information related to the general population, especially since no great differences are seen between the cohorts.

-        The footnote to the Table 1 should stick to the necessary information. It should not give methodological information: "EuroScore II: updated system for calculation the risk of death in heart surgery"; "Liver cirrhosis: cirrhosis of any stage according to Child-Pugh classification"; "Oral anticoagulant therapy: Vitamin K antagonists or new oral anticoagulants". In my opinion, this information should be included in the "Materials and methods" section.

-        Table 2: The title should be more concise: "Intraoperative data". I also think that in this table it is sufficient to provide comparative information between both cohorts, not giving the information related to the general population.

-        Table 3. The title should be more concise: “Postoperative outcomes”. The footnote to the Table 3 should not give methodological information: “Central neurological complications: ischemic and hemorrhagic events, encephalopathy, meningitis and brain abscess- transient or permanent”; “cumulative inotropes: (nor-)+epinephrine”; “deep surgical wound infection: including osteomyelitis or mediastinitis”; “total inotropes: (nor-) +epinephrine+dobutamine”; “urinary tract infection: including urethritis, cystitis and pyelonephritis”. This information should be included in the methodological section.

Discussion:

-        This section should start with the main findings, pointing them out in a more attractive way. If I am not mistaken that the true main objective of this analysis is to evaluate the effect of intraoperative hemoadsorption in reducing complications in patients with IE, I think that the authors should focus more on the effect of this device in reducing inflammation, postoperative complications and therefore mortality.

The Author Contributions and Funding sections should be filled in

Reviewer 2 Report

P9L257

"the study was not a randomized controlled trial, but both groups were comparable."

It would be good to check the absolute standardized difference (asd) whether the baseline of the retrospective study is comparable.
If asd differs between groups by more than a certain standard, comparison should be made after securing similarity between groups through propensity score matching.
In my opinion, although the number of the two groups is similar and the p value is not significant, it seems that the two groups cannot be said to be comparable in the present condition.

Table 4 is less readable.
It would be useful to separate sepsis related mortality from in hospital mortality.

Did you perfome logistic regression for sepsis itself?
